# Ethical Aspects of Brain Organoid Research in News Reports: An Exploratory Descriptive Analysis

**DOI:** 10.3390/medicina57060532

**Published:** 2021-05-27

**Authors:** Kazuki Ide, Norihiro Matsuoka, Misao Fujita

**Affiliations:** 1Division of Scientific Information and Public Policy, Center for Infectious Disease Education and Research (CiDER), Osaka University, Osaka 565-0871, Japan; 2Uehiro Research Division for iPS Cell Ethics, Center for iPS Cell Research and Application (CiRA), Kyoto University, Kyoto 606-8507, Japan; misao-fujita@cira.kyoto-u.ac.jp; 3The Institute of Natural Sciences, College of Humanities and Sciences, Nihon University, Tokyo 156-8550, Japan; 4Faculty of Medicine, Fujita Health University, Aichi 470-1192, Japan; norihiro.matsuoka@fujita-hu.ac.jp; 5Institute for the Advanced Study of Human Biology (ASHBi), Kyoto University, Kyoto 606-8501, Japan

**Keywords:** brain organoid, news report, database, ethics, moral, consciousness, self-awareness

## Abstract

*Background and Objectives*: Brain organoids are self-assembled, three-dimensional (3D) aggregates generated from pluripotent stem cells. These models are useful for experimental studies on human brain development and function and are therefore increasingly used for research worldwide. As their increasing use raises several ethical questions, we aimed to assess the current state of the press on brain organoid research using a cross-sectional database to understand the extent of discussion of this subject in the public. *Materials and Methods*: We conducted a descriptive analysis of news reports obtained from the Nexis Uni database, searched in April 2020. After extracting the news reports, the number of published reports in each year and the included terms were analyzed. *Results*: Up to April 2020, 332 news reports had been published, with over half of them published in the United States and the United Kingdom, with the numbers gradually increasing every year. In total, 113 (34.0%) news reports included ethics-related keywords, and the ratio of studies before and after the study-period midpoint was significantly increased (21.0% (2013–2016) vs. 38.2% (2017–2020); *p* = 0.0066, Chi-square test with Yates’ continuity correction). *Conclusions*: Although news reports on the ethical aspects of brain organoid research have been increasing gradually, there was a bias in the region of publication. Additional studies focusing on the ethical aspects of brain organoid research should strive to assess the public perception on the subject in different parts of the world.

## 1. Introduction

Brain organoids are self-assembled, three-dimensional (3D) aggregates generated from pluripotent stem cells such as human embryonic stem cells (hESCs) and human induced pluripotent stem cells (hiPSCs). These models are useful for experimental studies on human brain development and function [1,2]. Disease models based on brain organoids have also been established, such as organoids for idiopathic autism spectrum disorder (ASD) modeled by Mariani et al. [3] were established using iPSC lines from four patients with ASD, and an increased number of inhibitory interneurons was observed in the constructed brain organoids. One of the causes of this phenomenon was the upregulation of Forkhead Box G1 (*FOXG1*), which is involved in forebrain patterning. Other researchers revealed that a genetic condition causes smooth brain (lissencephaly) using a forebrain organoid model [4,5]. 

Trujillo et al. reported, in a 2018 preprint and in a 2019 peer-reviewed article, that the brain organoid models present electrical signal patterns that can be observed in premature babies [6,7]. The “In Focus News” section of Nature reported these findings when the preprint was published, and in the news article Prof. Koch of the Allen Institute for Brain Science in Seattle expressed “the ethical questions that this project raises about whether organoids could develop consciousness will be difficult to resolve” [8]. Nita A. Farahany and her colleagues also commented on the ethics of experiments with human brain models that get closer to replicating brain functions, even if we need to know more about “what consciousness is” in order to discuss that point [9]. In the commentary article, they stated “the possibility of organoids becoming conscious to some degree seems highly remote”. However, they mentioned the requirements of guidance and guidelines, and broader societal conversations [9]. 

Considering the opinions raised by researchers in the related fields and the growing trend of news reports for the general public, it is important to understand the extent of discussion of this subject in the public. Based on this background, we conducted a descriptive analysis to assess the current state of the press on brain organoid research using a cross-sectional database.

## 2. Materials and Methods 

### 2.1. Study Overview

The Nexis Uni (http://www.nexisuni.com (accessed on 20 May 2021); formerly LexisNexis Academic; New York, NY, USA) database was used to collect data for this study. A literature search was conducted in April 2020, and news reports related to brain organoid research were searched from 1975 (inception) to 2020, and relevant reports were extracted. A timeline of publication was then drawn and the keywords included in the news reports (details provided in the following section) were analyzed. 

### 2.2. Literature Search Strategy

News reports with keywords (“brain organoid” or “cerebral organoid”) were extracted. Based on the exploratory nature of this research, all literature extracted from the database was included. The data were organized as a dataset, with the following information as variables: (1) title, (2) year of publication, (3) publisher names, and (4) flags that indicate keywords related to the ethical aspects of brain organoid research (ethical/ethics, moral, conscious/consciousness, and self-aware/self-awareness). The publishers’ country was included as well, by searching the geographical information of the publishers’ office.

### 2.3. Primary and Secondary Outcomes

The primary outcome of this study was the number of published news reports in each year. The secondary outcomes were as follows: (1) distribution of the countries that published news reports, (2) the number of news reports that include keywords related to the ethical aspects of brain organoid research (ethical/ethics, moral, conscious/consciousness, and self-aware/self-awareness), and (3) trends in the appearance of each keyword in the news reports. The ratio of published news reports that included keywords before and after the midpoint year of the publication period (determined after the literature search process) was also compared.

### 2.4. Statistical Analysis

The data were descriptively analyzed; country names are expressed as number and percentage (%), and the timeline of publications is represented in the form of a bar graph, as the number of published news reports in each year. The ratio of published news reports that included keywords was compared using Chi-square test with and without Yates’ continuity correction, and both *p*-values obtained from the analysis were included to improve the accuracy of the analysis.

R version 3.6.1 for Windows (R Foundation for Statistical Computing, Vienna, Austria), JMP Pro 15 (SAS Institute, Cary, NC, USA), and Microsoft Excel 2019 (Microsoft Corporation, Redmond, WA, USA) were used for statistical analyses. Statistical significance was set at *p* < 0.05.

### 2.5. Ethical Consideration

The present study involved data from a cross-sectional database without any personal information. Therefore, ethical approval by the institutional review board was waived.

## 3. Results

Literature search of the database resulted in inclusion of 332 news reports related to brain organoid research. The timeline of the publication is demonstrated in Figure 1. The first report was published in 2013, when Lancaster et al. reported the establishment of human pluripotent stem cell-derived 3D cerebral organoids [1], and the number of related news reports has gradually increased every year since then.

The countries with published news reports are listed in Table 1. Of the news reports published, 38.6% (*n* = 128) were from the United States and 28.6% (*n* = 95) were from the United Kingdom. Other countries accounted for less than 10% of the published news reports.

The total number of news reports that included ethics-related keywords (ethical/ethics, moral, conscious/consciousness, and/or self-aware/self-awareness) was 113 (34.0%). The year 2016 was the midpoint of the period when the news reports were published; therefore, we compared the ratio of published news reports that included the ethics-related keywords, before and after the midpoint according to the pre-determined secondary outcome of this study. The ratio increased from 21.0% (2013–2016) to 38.2% (2017–2020), and this increase was statistically significant (*p*-value without Yates’ continuity correction = 0.0044, *p*-value with Yates’ continuity correction = 0.0066; Table 2).

## 4. Discussion

In this study, we referred to a news report database and analyzed the timeline of publications related to brain organoid research. Since 2013 (the year of establishment of a human pluripotent stem cell-derived 3D cerebral organoid model), 332 news reports have been published, and the number has increased gradually. This is the first study to evaluate the ethical aspects of brain organoid research mentioned in news reports that can provide useful information to understand the changes in social interest/perspective regarding this topic over time.

The United States and the United Kingdom accounted for more than half of all published news reports. This can be attributed to the fact that the language of search was limited to English. Nevertheless, there were a few news reports from Australia and other English-speaking countries; however, it is important to note this disparity.

The number of news reports was three times higher during 2017–2020 than during 2013–2016, with a particularly high number of published reports in 2018 and 2019 compared with that in the other years (over 80 reports in both years). This indicates a public focus on the ethical aspects of brain organoid research, and an increase in the level of attention, especially in the last few years. When focusing on the discussion on ethics by researchers in related fields, Farahany et al. stated “the possibility of organoids becoming conscious to some degree seems highly remote” as we cited in the Introduction. However, they also stated “the closer the proxy gets to a functioning human brain, the more ethically problematic it becomes” [9]. Other researchers have also stated that conscious self-awareness may not be a serious ethical challenge for biomedical research [10]. Researchers have also expressed concerns regarding the overemphasis on organoid consciousness and moral status, with these opinions based on the current progression of the related research fields [11]. These opinions may be acceptable by researchers and individuals well informed about the current stage of brain organoid research and brain sciences; however, it is not clear how the general public perceives it. When brain organoid research first came to the public’s attention, the public image of such studies may have diverged from researchers’ perceptions, partially because of news reports linking it to Frankenstein [12]. For this reason, it is essential to conduct surveys to understand the opinions of the general public to clarify public perception. Furthermore, when such surveys are performed, it should be taken into consideration whether or not the community has a perception of the issue.

A limitation of the current study is that we focused only on the number of news reports, their timeline, and the presence of keywords related to the ethical aspects of brain organoid research. Therefore, it is difficult to know the details of each news report. These keywords can be used in other contexts. Furthermore, the reports may also represent potentially lurid interpretations of the research chosen to increase readership. However, this study is important for understanding the current position of the society on the topic, and may also serve as a basis for future research focusing on the contents of news reports. The results are also useful for comparison purposes when researchers conduct a study focusing on published scientific reports.

Hyun et al. organized the following five possible themes associated with the ethics of brain organoids: (1) research oversight, (2) human biomaterial procurement and donor consent, (3) translational delivery, (4) animal research, and (5) organoid consciousness and moral status. These points may be useful to classify news reports on brain organoid research. In addition, the topics would be useful for surveys and analysis of public understanding of brain organoid research, in order to shed light on their concerns in future investigations.

## 5. Conclusions

In conclusion, there has been an increase in the discussion on the ethical aspects of brain organoid research in public forums since 2018. However, the number of reports is still limited, and further discussion with stakeholders may be needed along with technological advancements. Additional studies focusing on the ethical aspects of brain organoid research should strive to assess the public perception on the subject in different parts of the world.

## Figures and Tables

**Figure 1 medicina-57-00532-f001:**
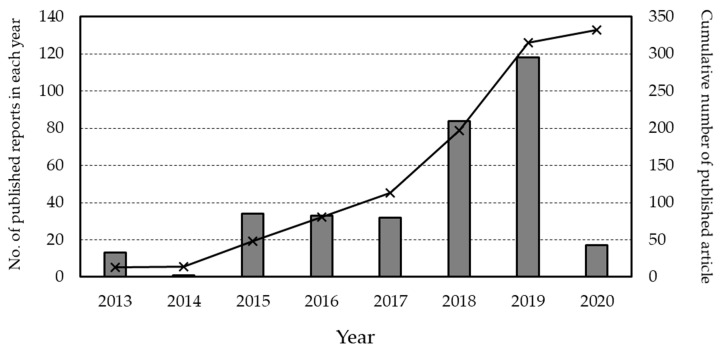
Number of new reports related to brain organoid research from 2013 to 2020. The bar graph shows the number of published reports in each year and the line graph represents the cumulative number of published reports. As 2013 is the first year of brain-organoid research-related news report publication, the timeline after that year has been drawn. The number of published articles in 2020 is included up to April.

**Table 1 medicina-57-00532-t001:** Countries with brain organoid research-related published news reports.

Rank ^1^	Country	*n*	%
1	The United States	128	38.6
2	The United Kingdom	95	28.6
3	India	24	7.2
4	The Netherlands	13	3.9
5	Australia	10	3.0
6	Austria	8	2.4
7	Germany	7	2.1
8	Canada	4	1.2
9	Swiss	4	1.2
10	Armenia	3	0.9
10	Iran	3	0.9
10	Thailand	3	0.9
	Other Countries ^2^	30	
Total		332	100

^1^ Rank based on the number of published news reports in each country. ^2^ The following countries are included in the category: Ireland, Singapore, South Africa, Bahrain, Spain, France, Israel, Kenya, Pakistan, and unknown.

**Table 2 medicina-57-00532-t002:** Number of news reports that included ethics-related keywords.

Time Period	Included	Not Included	*p*-Value ^1^	Corrected *p*-Value ^1^
Year	*n* (%)	*n* (%)		
2013–2016 (*n* = 81)	17 (21.0)	64 (79.0)	0.0044	0.0066
2017–2020 (*n* = 251)	96 (38.2)	155 (61.8)		
Total	113	219	–	–

^1^ *p*-value was calculated using Chi-square test, with and without Yates’ continuity correction.

## Data Availability

Owing to the conditions and usage of the database used in this study, the data are not publicly available. However, the database is commercially available, and the data can be extracted from Nexis Uni (http://www.nexisuni.com (accessed on 20 May 2021)).

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
