# Peer review of "Ethical Aspects of Brain Organoid Research in News Reports: An Exploratory Descriptive Analysis"

_medicina, 2021, doi:10.3390/medicina57060532_

Round 1

Reviewer 1 Report

I still feel that the overall premise of this research is of limited value.It could well be saying more about what stories get picked up by the media for "click bait" (and the Frankenstein allusion suggests this). However the qualifications included have improved that a bit

Author Response

We thank you for your suggestions and constructive feedback. We will also try our best to conduct more useful investigations in the near future.

Reviewer 2 Report

My concerns have been addressed, if just minimally. Only my last comment has not been addressed sufficiently: Item 12 in the reference section is still incomplete (missing name of author, as well as date and place of publication). I provided all the data in my previous review and it is clearly stated on the website.

Author Response

We thank you for your constructive feedback and we apologize for the inconvenience. We have updated the reference according to the information which provided by you.

>Reference: Kreeger K. Mini-organs: Next-gen lab model, not the child of Frankenstein. Penn Medicine News Blog (posted on October 25, 2018). Available online: https://bit.ly/3tLd4vc (accessed on 20 May 2021).